# Relationship between Overweight/Obesity and Social Communication in Autism Spectrum Disorder Children: Mediating Effect of Gray Matter Volume

**DOI:** 10.3390/brainsci13020180

**Published:** 2023-01-21

**Authors:** Wei Cheng, Zhiyuan Sun, Kelong Cai, Jingjing Wu, Xiaoxiao Dong, Zhimei Liu, Yifan Shi, Sixin Yang, Weike Zhang, Aiguo Chen

**Affiliations:** 1College of Physical Education, Yangzhou University, Yangzhou 225127, China; 2Institute of Sports, Exercise and Brain, Yangzhou University, Yangzhou 225127, China

**Keywords:** autism spectrum disorder, gray matter volume, mediating effect, overweight/obesity, social communication

## Abstract

With advances in medical diagnostic technology, the healthy development of children with autism spectrum disorder (ASD) is receiving more and more attention. In this article, the mediating effect of brain gray matter volume (GMV) between overweight/obesity and social communication (SC) was investigated through the analysis of the relationship between overweight/obesity and SC in autism spectrum disorder children. In total, 101 children with ASD aged 3–12 years were recruited from three special educational centers (Yangzhou, China). Overweight/obesity in children with ASD was indicated by their body mass index (BMI); the Social Responsiveness Scale, Second Edition (SRS-2) was used to assess their social interaction ability, and structural Magnetic Resonance Imaging (sMRI) was used to measure GMV. A mediation model was constructed using the Process plug-in to analyze the mediating effect of GMV between overweight/obesity and SC in children with ASD. The results revealed that: overweight/obesity positively correlated with SRS-2 total points (*p* = 0.01); gray matter volume in the left dorsolateral superior frontal gyrus (Frontal_Sup_L GMV) negatively correlated with SRS-2 total points (*p* = 0.001); and overweight/obesity negatively correlated with Frontal_Sup_L GMV (*p* = 0.001). The Frontal_Sup_L GMV played a partial mediating role in the relationship between overweight/obesity and SC, accounting for 36.6% of total effect values. These findings indicate the significant positive correlation between overweight/obesity and SC; GMV in the left dorsolateral superior frontal gyrus plays a mediating role in the relationship between overweight/obesity and SC. The study may provide new evidence toward comprehensively revealing the overweight/obesity and SC relationship.

## 1. Introduction

Autism spectrum disorder (ASD) is a lifelong, severe psychoneurotic developmental disorder occurring in early childhood that limits or impairs daily functioning [1]. One of the core symptoms of ASD is social communication (SC) impairment. Children with ASD generally show SC impairments in understanding, maintaining, and developing social relationships [2]. It severely hinders them from making friends, and leads to a consequent decline in physical activity and an increase in sedentary time [3]. Notably, prolonged sedentary time can induce anxiety or obesity, and can lower quality of life. Additionally, negatively impacting SC, obese children are vulnerable to bullying from peers [4,5]. A total of 1490 obese children aged 7–12 were surveyed on their health condition, and it was found that obese children suffer from severe depression [6]. The investigation of the dietary habits of 54 children with ASD presented a 42.6% obesity rate among children with ASD and discovered a positive correlation between obesity and symptom severity [7]. Through the study of children with ASD aged 3–7 years, Yang et al. found that the severity of symptoms can be accompanied by severe dietary behaviors that lead to a rapid increase in an individual’s BMI [8]. Based on this, they concluded that social impairment in children with ASD may strongly associate with a rapid increase in BMI. By examining the prevalence and health correlations of overweight/obesity in children with ASD, scholars have found that overweight/obese children with ASD participate in fewer physical activities and skill developments, leading to poor physical coordination and weak communication skills [9].

Generally, many pieces of research found that overweight/obese children with ASD may also exhibit a range of abnormal responses, including social isolation or aggressive behavior. Notably, a study in 2021 on obesity genes and social functioning in ASD confirmed a link between obesity genes and social functioning [10]. However, previous studies have only suggested that overweight/obesity may be a potential factor influencing social impairment, and that the relationship between them has not been confirmed. Therefore, the first hypothesis in this study is proposed: there may be a significant correlation between overweight/obesity and SC in children with ASD.

Up until now, the neural mechanisms affecting SC in children with ASD have become a hot research spot. Social communication in children with ASD can be affected by different mechanisms, which are mostly related to neural mechanisms [11,12,13]. Typically, gray matter volume (GMV) has been widely studied as a neural mechanism. The “social brain” theory of brothers revealed that brain regions are closely related to social functioning, where reduced GMV in children with ASD is the key to SC impairment in children with ASD [14]. Growing evidence demonstrates abnormal GMV in several brain regions in ASD patients, including the frontal, temporal, parietal, anterior cingulate, insula, caudate nucleus, and lingual gyrus [15,16,17]. During the growth of children with ASD, lower GMV has been observed in the inferior and superior frontal gyri, medial and superior occipital gyri, hippocampus, and the left hemisphere of the epencephalon, frequently leading to SC impairment [18,19]. There is a strong association between SC impairment and reduced GMV in the frontal lobe, frontal–trigeminal cerebellar region, and superior temporal sulcus in children with ASD [20,21]. GMV has a strong correlation with SC in children with ASD. However, whether GMV is a neural mechanism between overweight/obesity and SC in children with ASD has not been revealed.

Research in the field of brain science has concluded that being overweight/obese causes poor brain health [22], such as reduced gray matter and white matter. The World Health Organization and the National Center for Health Statistics apply BMI as an indicator to define overweight/obesity. Research evidence reveals that children with ASD are at greater risk for obesity [23]. It was illustrated that obesity not only poses a threat to the growth and nutrition of children [24], but also contributes to numerous diseases such as hypertension [25], diabetes [26], and many other dangerous diseases [27]. A study in 2021 found a significant negative association between overweight/obesity and GMV [28]. A 1-year follow-up of adolescents demonstrated that the increase in BMI is associated with a decrease in frontal GMV [29]. The same study explored the relationship between childhood obesity and GMV, indicating that changes in GMV in the pallidum [30,31], thalamus, cerebellum, amygdala [32], cingulate [33], and hippocampus [34] regions are associated with obesity in children. Moreover, a review found that an elevated BMI in adolescents is associated with reduced GMV, mainly in prefrontal and limbic regions [35].

In summary, overweight/obesity is significantly negatively correlated with GMV. However, the relationship between overweight/obesity and GMV among children with ASD has not been conclusively established.

With the development of brain imaging technology, brain plasticity has been confirmed. Currently, correlational studies hotspot the idea to utilize GMV as a mediator in the methodology while the relationship between overweight/obesity, GMV, and SC in children with ASD has not been fully revealed, especially considering whether or not GMV is a potential neural mechanism for overweight/obesity and SC. Therefore, hypothesis two in this study is proposed: there is a mediating role of GMV in this relationship between overweight/obesity and SC.

## 2. Methods

### 2.1. Research Subjects

In this study, we recruited 147 children aged 3 to 12 years old who were diagnosed with ASD according to the DSM-5 criteria from the Chuying Child Development Center, Starssailor Educational Institution, and Maternal and Child Health Hospital (Yangzhou, China). Only 101 participants completed the study (Figure 1).

Inclusion criteria: (1) Han ethnicity; (2) Aged 3 to 12 years old; (3) Use of the Subject Checklist (V2.0) of the Brain Imaging Center of the Affiliated Hospital of Yangzhou University to check if the conditions for sMRI scanning are satisfied, including the absence of metal implants (such as metallic dentures) and electronic, magnetic, or mechanical devices (such as pacemakers).

Exclusion criteria: (1) Head injury; (2) Nervous system disorders such as phenylketonuria, epilepsy, tics, mental illness, etc.; (3) Impaired hearing and vision; (4) Use of drugs that affect the central nervous system.

After further screening and exclusion based on experimental criteria, thirty-five children were excluded due to the above-mentioned reasons or declined to participate. Additionally, it was found that after the collection of behavioral and brain data, head movement in 7 children’s sMRI data was greater than 3mm, and 4 children had not undergone an sMRI scan. In total, 101 children with ASD were included in the study, of which 89 were male (88.12%) and 12 female (11.88%). The parents or guardians of all children who participated in this trial have signed an informed consent form according to the provision of the Declaration of Helsinki. The study was approved by the Ethics and Human Protection Committee of the Affiliated Hospital of Yangzhou University and was registered (ChiCTR190002497) in the Chinese Clinical Trials Registry.

### 2.2. Behavioral Measurements

All participants’ height and weight were measured using the Meilen Smart Height and Weight Scale (Model: MSG005-H), Manufacturer: Shenzhen Mobil Electronics Wired Company. The error range of the experimental instrument is 0.5 cm and 0.1 kg, which can automatically save data and ensure the accuracy of the test data. Two disciplined physical education staff were authorized to ensure the consistency of the experimental testers during the test, and all subjects were requested to wear light clothing and undress their shoes and socks during the measurement. BMI was calculated using the traditional calculation method BMI = (weight (kg))/(height (m))^2^. The overweight/obesity scoring criteria thresholds were based on the 2010 Chinese 2–18 years BMI values of children and adolescents [36]. Among the participants, 13 children were overweight (12 boys/1 girls) making 12.87%, and 20 children were obese (18 boys/2 girls) making 19.80%.

The Social Responsiveness Scale, Second Edition (SRS-2) was used to evaluate the social skills of children with ASD. This scale was developed by John N. Constantino and Christian P. Gruber in 2012 to assess the social skills of individuals with autism from 2 years and 6 months to adulthood [37]. The scale consists of 65 questions divided into 4 subscales. Chinese scholars performed a reliability check [38], yielding a Cronbach’s αcoefficient of 0.946 for the SRS-2’s overall scale, with dimensions between 0.563 and 0.902; the half-confidence was 0.958, with dimensions between 0.710 and 0.882. The scale is scored as a whole, with higher scores indicating more severe symptoms of SC. The scale can be easily applied to assess the social interaction of the participants. Parents or guardians of all subjects completed the questionnaire according to their daily performance.

### 2.3. Collection and Treatment of Gray Matter Volume

This study was carried out at the Affiliated Hospital of Yangzhou University. Due to the duration and noise of the sMRI scan, the subjects were subjected to moderate sleep deprivation to make sure all the children with ASD were able to participate in the scan. The day before the examination, parents were informed to let the subjects sleep late and wake up early. After an interval of 6–8 h, each subject was given 10% chloral hydrate 0.3 mL/kg (30 mg/kg) by enema, without exceeding a maximum dose of 10 mL. Once the subjects fell asleep, the nurse tested their level of awareness. If the patient did not respond to mild pain stimuli, the patient was placed in the supine position and sMRI was performed.

Image acquisition was completed using the GE Discovery MR750W 3.0T MRI. The t1-MPRAGE structural image scan parameters were: pulse repetition interval = 7.20 ms, echo time = 3.06 ms, thickness = 1.00 mm, flip angle = 12°, acquisition matrix = 256 × 256, and scan field of view = 256 × 256 mm. Based on the MATLAB (2018) platform, the CAT12 toolbox (<Introduction for CAT12 Toolbox> CAT (https://neuro-jena.github.io/) (accessed on 28 March 2022)) that is based on SPM (<Introduction for SPM Software> https://www.fil.ion.ucl.ac.uk/spm/ (accessed on 28 March 2022)) was applied to automate the preprocessing of T1-MPRAGE data. The preprocessed data were smoothed in SPM12 with a gaussian kernel function of 8 mm × 8 mm × 8 mm half-width full-height to improve the image signal-to-noise ratio. Referring to the gray matter volume extraction method of Aiguo Chen et al. (2018) [39], the get_totals plug-in was used to extract the values of the GMVs of 116 brain regions in the whole brain.

### 2.4. Statistical Analysis

SPSS22.0 (SPSS; SPSS Inc., Chicago, IL, USA) was employed for all statistical analyses, and the statistical significance was defined as *p* < 0.05. The partial correlation analysis was then used to examine the relationship between overweight/obesity and the GMVs of the relevant brain regions. The correlations between GMV, scores on the overweight/obesity, and SC scales were then analyzed. Partial correlation analysis was used to control for age and gender and to assess the correlation between overweight/obesity, GMV (brain regions that differed significantly between groups), and SC. In addition, a mediation analysis was performed taking overweight/obesity as a predictor (X) and GMV and SC as outcome variables (Y). Model 4 of the Process plug-in in SPSS was applied to test the indirect influence of overweight/obesity on SC directly with GMV. Standardized (z-trans) formed variables were used in the mediation analysis. The bias-corrected percentile Bootstrap method was used, and the 95% confidence interval of the mediating effect was estimated from 5000 Bootstrap samples. If the confidence interval did not include 0, the indirect effect of the mediating model was considered statistically significant.

## 3. Results

### 3.1. Correlation Analysis between Overweight/Obesity, GMV, and SC in Children with ASD

The Kolmogorov–Smirnov test was utilized to assess normal distribution in BMI, GMV, and SC. It fits the normal distribution. Partial correlation analysis with gender and age as a controlled variable was applied to evaluate dimensional correlations where *p* < 0.05 was considered significant (Table 1). Demographic characteristics of each group are presented as means (M) and standard deviations (SD).

The results (Table 2) revealed a significant positive correlation between overweight/obesity and SRS-2 total points (*r* = 0.343, *p =* 0.01) in children with ASD (Figure 2a). The brain regions in which GMV was associated with SC included the left middle temporal gyrus (*r* = −0.227, *p* = 0.024), Frontal_Sup_L (*r* = −0.432, *p* = 0.001), and right hippocampus (*r* = −0.243, *p* = 0.015). The brain areas associated with overweight/obesity and GMV included the right amygdala (*r* = 0.224, *p =* 0.026), Frontal_Sup_L (*r* = −0.354, *p =* 0.001), and the right thalamus (*r* = 0.238, *p* = 0.018). Only the gray matter volume in the left dorsolateral superior frontal gyrus (Frontal_Sup_L GMV) was correlated with both SC and overweight/obesity in children with ASD (Figure 2b,c). Our results validate research hypothesis one and are similar to the results of previous studies [6,7,11,12,13,23].

### 3.2. Mediation Test of GMV between Overweight/Obesity and SC

Our study found a correlation between overweight/obesity, Frontal_Sup_L GMV, and SC. To further explore the interrelationship between the three evaluated parameters, the approach of Preacher and Hayes was employed to test the saliency of the indirect effects. A mediating effect analysis was conducted using Model 4 (simple mediation model) of the Process plug-in in the SPSS (SPSS; SPSS Inc., Chicago, IL, USA) software package, with overweight/obesity as the independent variable. In this analysis, gender and age, Frontal_Sup_L GMV, and SC were set as the control variable, mediating variable, and dependent variable, respectively. Results of the regression analysis (Table 2) showed that overweight/obesity (*B* = 0.352, *t* = 3.598, *p* = 0.01) positively predicted SC in children with ASD in Model 1. Meanwhile, overweight/obesity (*B* = −0.354, *t* = −3.723, *p* = 0.001) was found to negatively predict GMV in the left dorsolateral superior frontal gyrus in children with ASD in Model 2. In Model 3, overweight/obesity (*B* = 0.223 *t* = 2.270, *p* = 0.001) and GMV in the left dorsolateral superior frontal gyrus (*B* = −0.363, *t* = −3.704, *p* = 0.001) jointly predicted SC in children with ASD.

In the mediation effect test, the Bootstrap method was selected in the mediation model (Table 3). The upper and lower limits Bootstrap to 95% confidence intervals for the direct effects of overweight/obesity on SC, and the mediating effects of Frontal_Sup_L GMV did not contain 0, implying that overweight/obesity can predict SC directly or through the mediating effects of Frontal_Sup_L GMV (Figure 3). The direct effect value (0.223) and mediating effect value (0.129) accounted for 63.4% and 36.6% of the total effect value (0.352), respectively.

## 4. Discussion

In this study, the integration of behavioral and imaging techniques revealed that overweight/obesity in children with ASD is closely associated with SC. The GMV of the left dorsolateral superior frontal gyrus was a significant mediator in the link between overweight/obesity and SC, which validated our hypothesis. Higher BMI’s in children with ASD is associated with higher SC scores, indicating that overweight/obesity in children with ASD has a significant positive predictive effect on SC. The direct effect size was 63.4%. Regarding previous studies on overweight/obesity and social impairment in children with ASD, it was found that overweight/obesity in children with ASD was associated with reduced dorsolateral superior frontal gyrus GMV, and social impairment in children with ASD was associated with reduced dorsolateral superior frontal gyrus GMV. Therefore, the relationship between overweight/obesity and social impairment in children with ASD could be modulated by increasing the dorsolateral superior frontal gyrus GMV. Significantly, the mediating model was applied to explore the mediating role of the GMV of the left dorsolateral superior frontal gyrus in the relationship between overweight/obesity and SC. It was found that potential neural pathways were involved in the association, which provides new evidence to comprehensively uncover the relationship between overweight/obesity, GMV, and SC.

### 4.1. Correlation between Overweight/Obesity, GMV, and SC in Children with ASD

A positive association between overweight/obesity in children with ASD and SC impairment was found, indicating that a higher BMI in children with ASD resulted in an increased risk of catching SC. The results are consistent with hypothesis one, and provide preliminary confirmation of the association between overweight/obesity in children with ASD and SC. In a previous study in islanders, children with overweight/obesity and ADHD were found to have severe social impairment [40]. Additionally, in a systematic review of overweight/obesity in children with autism, a high BMI in children with ASD was found to be significantly and positively associated with their core symptoms [41]. Based on the analysis, it can be concluded that there are numerous associations between overweight/obesity and SC in children with ASD. About 60% of children with ASD have hormonal heterogeneity due to the long-term use of antipsychotic medications [42,43]. Furthermore, they have food selectivity and special dietary habits, which lead to an abnormally high BMI and overweight/obesity [44,45]. Overweight/obesity in children with ASD is associated with increased sedentary time and decreased physical activity. The Chinese Childhood Obesity Report shows that overweight/obese children have extremely low self-esteem and are reluctant to engage with others. This leads to a reduction in their time spent interacting with peers. With less contact with the outside world, children with ASD will avoid interacting with people [46,47], resulting in increased social damage similar to our findings. Therefore, to prevent the rapid increase in BMI in children with ASD, targeted programs should be launched to enhance the social skills of children with ASD. Additionally, it can be salutary to improve their dietary habits and increase their participation in physical activities. For example, with the help of machine therapy, highly sensitive children with ASD can improve their social interactions and reduce sedentary time [48].

In this study, a significant negative correlation was found between the GMV of the left dorsolateral superior frontal gyrus and SC in children with ASD. Previous studies showed a significant relationship between GMV and SC in patients with ASD, specifically in the frontal and temporal regions [49], the superior frontal gyrus, and the middle temporal gyrus [50,51]. Studies on the social brain network in adults with ASD revealed that reduced GMV in several brain regions, such as the right inferior occipital gyrus, left syrinx gyrus, right middle temporal gyrus, bilateral amygdala, right inferior frontal gyrus, right orbitofrontal lobe, and left dorsomedial prefrontal lobe, impacted their SC [52,53]. The result in this article is consistent with existing studies, and our study shows that the GMV of the left dorsolateral superior frontal gyrus is also one of the neural mechanisms affecting SC, where numerous findings also suggest that the GMV of the left dorsolateral superior frontal gyrus plays an important role in the brain. Firstly, it has been suggested that the dorsolateral superior frontal gyrus is closely related to emotion regulation, cognitive processes, and self-awareness [54], providing a higher neural basis for cognitive function. The dorsolateral superior frontal gyrus may also be crucial to SC in children with ASD. Secondly, the left dorsolateral superior frontal gyrus is an important brain component area of the default mode network (DMN), and DMN abnormalities are closely related to social interaction and verbal communication impairment in ASD [55]. In addition, the activity of the DMN is closely related to the memory, emotion, and cognitive functions of the human brain. The reduced GMV of the left dorsolateral superior frontal gyrus impairs the normal activity of the DMN [56,57,58], which may lead to poorer SC in children with ASD. Findings from previous studies have discovered a strong association between reduced GMV in the inferior frontal gyrus, amygdala, and thalamus and social impairment in children with ASD. The study in this article did not find that strong association, probably attributed to the discrepancy of the subjects (the subjects in this article were younger children, but their studies were predominantly adults) [59]. Based on the analysis above, the present study also verified that the GMV of the left dorsolateral superior frontal gyrus significantly affects SC in children with ASD through neural mechanisms.

In this study, there was a negative correlation between overweight/obese children with ASD and the GMV of the left dorsolateral superior frontal gyrus. The rise in the BMI in children with ASD may be associated with changes in the GMV of the left dorsolateral superior frontal gyrus. In the field of brain research, many studies have shown that BMI is inversely associated with GMV, although regional patterns often differ across studies [60,61]. Furthermore, after participating in physical activity, obese subjects had a significant decrease in BMI and a significant increase in gray matter in brain regions [62,63]. However, some studies disagree with the findings of this study. Obese individuals are associated with reduced GMV in the middle temporal lobe, prefrontal cortex, hippocampus, and amygdala [64]. Our study found no association between these brain regions, possibly because our study group was children with ASD. There could be many reasons for this result. Studies have shown that a higher BMI leads to decreased neuronal and myelin viability [65], while damage from neuron proliferation or excessive dendritic pruning also leads to reduced GMV [66]. At the same time, studies have shown that high BMI can lead to the thickening of blood vessel walls or dysfunction of vascular endothelium, damage to carotid arteries, reduced blood supply to the brain, and lower GMV [67]. In summary, high BMI causes structural brain abnormalities and also affects brain structures. Monitoring the lifestyle of overweight/obese children with ASD to promote a balanced diet and regular physical activity to optimize BMI levels may reduce the risk of abnormal GMV.

### 4.2. Mediating Effect of GMV in the Relationship between Overweight/Obesity and SC in Children with ASD

This study further employed PROCESS to establish the mediating effect based on correlations among overweight/obesity, GMV, and SC. It was established that the Frontal_Sup_L GMV of children with ASD played a partial mediating role between overweight/obesity and SC. Higher BMI’s resulted in smaller Frontal_Sup_L GMV, thereby affecting neural connections in areas such as the frontal lobe. Thus, the activity of the dorsolateral superior frontal gyrus neural network is reduced. This may make social barriers to ASD elevated. Firstly, previous studies have shown that overweight/obesity may lead to structural and functional brain abnormalities in children with ASD, such as abnormally reduced GMV in the frontal and superior temporal sulci [64], which may cause SC impairment in children with ASD. Secondly, GMV is closely related to structural brain function, and some studies have shown that abnormal GMV in the frontal, temporal, and amygdala brain regions may be an important manifestation of SC in children with ASD [49,68,69]. Overweight/obesity is a potential risk factor for structural brain alterations in ASD, and numerous studies have shown that the physical health and SC of children with ASD can be improved by physical interventions (horseback riding, recreational games, etc.) [70,71]. Finally, the physical health of children with ASD significantly improved after a 12-week mini-basketball intervention, along with an increase in GMV in brain regions and enhanced social skills [72,73]. The results of this study showed an indirect effect size of 36.6% of GMV in the left dorsolateral superior frontal gyrus between overweight/obesity and SC. It is reasonable to assume that the GMV of the left dorsolateral superior frontal gyrus may be one of the multiple neural mechanisms. For example, white matter, cortex, and other neural mechanisms have yet to be uncovered. The work in the article may also provide evidence for the mediating role of GMV in the relationship between overweight/obesity and SC in children with ASD.

### 4.3. Research Limitation and Outlook

This study expands on previous studies that primarily preliminarily confirmed the association between overweight/obesity and SC in children with ASD, and reveals that GMV may be one of the neural mechanisms of the association between overweight/obesity and SC. However, the present study also has some limitations. First, the sample size of overweight/obese children with ASD was relatively small. The study population included children with ASD, which involved difficulties in recruiting participants. Second, this paper is mainly a preliminary exploration of the whole-brain neural mechanisms between overweight/obesity and social impairment, without analysis of brain regions of interest, and in the future, analysis of interest in both can be considered. Third, this study is a cross-sectional study. Longitudinal interventions can be used in future studies to further confirm the neural mechanisms of this study. The factors affecting SC are complex and diverse, and subsequent studies can consider the effects of other factors such as molecular and cellular levels.

## 5. Conclusions

A positive association between overweight/obesity and SC was observed in children with ASD. At the same time, in children with ASD, a negative association was found between overweight/obesity and the GMV of the left dorsolateral superior frontal gyrus, and a negative association between SC and the GMV of the left dorsolateral superior frontal gyrus. The GMV of the left dorsolateral superior frontal gyrus mediated the association between overweight/obesity and SC. In conclusion, we have demonstrated that GMV is one of the neural mechanisms underlying the association between overweight/obesity and social impairment. Moreover, findings in this article may have some public health implications by giving suggestions to overweight/obese children with ASD who are vulnerable to social impairment in the future. For instance, by participating in physical activity, children with ASD who are overweight/obese can reduce their BMI and be effective in slowing down abnormalities in brain GMV, thereby improving their social interaction skills.

## Figures and Tables

**Figure 1 brainsci-13-00180-f001:**
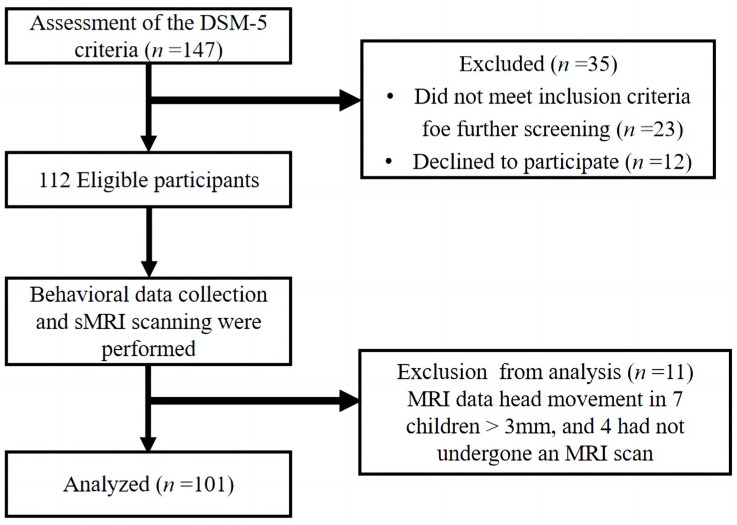
Experimental flow chart.

**Figure 2 brainsci-13-00180-f002:**
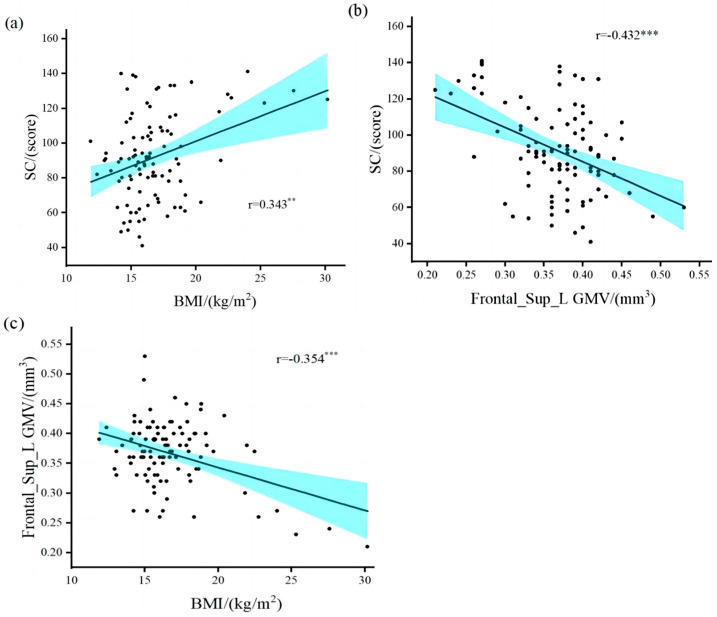
Correlation between overweight/obesity, gray matter volume, and social communication. (**a**) overweight/obesity was positively correlated with SRS-2 total points; (**b**) gray matter volume in the left dorsolateral superior frontal gyrus was negatively correlated with SRS-2 total points; (**c**) overweight/obesity was negatively correlated with gray matter volume in the left dorsolateral superior frontal gyrus. BMI, Body Mass Index; Frontal_Sup_L GMV, gray matter volume in the left dorsolateral superior frontal gyrus; SC, Social Communication. ** Denotes statistical significance at *p* < 0.01. *** Denotes statistical significance at *p* < 0.001.

**Figure 3 brainsci-13-00180-f003:**
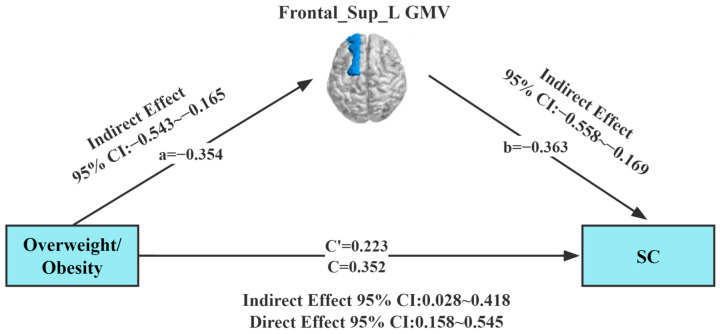
Model of the mediating effect of gray matter volume in the relationship between overweight/obesity and social communication. All variables in the model are substituted into the regression equation by the standardized variables. Frontal_Sup_L GMV (dark blue area), gray matter volume of the left dorsolateral superior frontal gyrus; SC, Social Communication.

**Table 1 brainsci-13-00180-t001:** The means of each study variable, standard deviations, and correlation coefficient between variables.

	M ± SD	BMI	Frontal_Sup_L GMV	SC
BMI	16.78 ± 2.91			
Frontal_Sup_L GMV	0.368 ± 0.055	−0.354 ***		
SC	91.78 ± 24.65	0.343 **	−0.432 ***	
Overweight	13	12	1	
Obesity	20	18	2	

Note: A total sample size of n = 101 was included in the analysis. Descriptive data are presented as means (M) and standard deviations (SD). BMI, Body Mass Index; Frontal_Sup_L GMV, gray matter volume in the left dorsolateral superior frontal gyrus; SC, Social Communication. ** Denotes statistical significance at *p* < 0.01. *** Denotes statistical significance at *p* < 0.001.

**Table 2 brainsci-13-00180-t002:** Regression analysis of the relationship between variables in the model of the mediating effect of GMV.

Predictive Variables	Model 1		Model 2		Model 3	
	*B*	*t*	*B*	*t*	*B*	*t*
BMI	0.352	3.598 **	−0.354	−3.723 ***	0.223	2.270 ***
Frontal_Sup_L GMV					−0.363	−3.704 ***
*R^2^*	0.351		0.412		0.483	
*R-sq*	0.123		0.170		0.323	
*F*	4.541 **		6.610 ***		7.280 ***	

Note: All variables in the model are substituted into the regression equation by the standardized variables. Model 1, overweight/obesity predicts social communication; Model 2, overweight/obesity predicts gray matter volume; Model 3, overweight/obesity and gray matter volume jointly predict social communication. BMI, Body Mass Index; Frontal_Sup_L GMV, gray matter volume in the left dorsolateral superior frontal gyrus; SC, Social Communication; ** Denotes statistical significance at *p* < 0.01. *** Denotes statistical significance at *p* < 0.001.

**Table 3 brainsci-13-00180-t003:** Mediating effect of GMV between overweight/obesity and SC.

	Effect	Boot SE	Boot LLCI	Boot ULCI	Proportion
Direct effect	0.223	0.098	0.028	0.418	63.4%
Indirect effect	0.129	0.049	0.026	0.220	36.6%
Total effect	0.352	0.098	0.158	0.545	100%

Note: All variables in the model are substituted into the regression equation by the standardized variables. SE, standard error; LLCI and ULCI, lower and upper levels for the confidence interval. Boot SE, Boot LLCI, and Boot ULCI estimate the standard error of indirect effect and the lower and upper limit of 95% confidence interval using the percentile bootstrap method with deviation correction, respectively.

## Data Availability

The data presented in this study are available on request from the corresponding author. The data are not publicly available due to the privacy of participants.

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
