# Peer review of "Relationship between Overweight/Obesity and Social Communication in Autism Spectrum Disorder Children: Mediating Effect of Gray Matter Volume"

_brainsci, 2023, doi:10.3390/brainsci13020180_

Round 1
Reviewer 1 Report
Comments and Suggestions for Authors
1. As your abstract's final sentence, include a "take-home" message.
2. Rearrange keywords alphabetically.
3. Nothing truly unique in its current state. Because of the lack of novel, the current study looks to be a replication or modified study. The authors must describe their novel in detail. This work should be rejected owing to a major issue.
4. Previous study related needs to explain in the introduction section consisting of their work, their novelty, and their limitations to show the research gaps that intend to be filled in the present study.
5. The authors need to explain some treatment in children with autism, one of the with sensory hug machine. It is a vital topic that authors must provide in the introduction and/or discussion section. Additionally, the MDPI's suggested reverence should be taken to substantiate this explanation as follows: Afif, I. Y.; Manik, A. R.; Munthe, K.; Maula, M. I.; Ammarullah, M. I.; Jamari, J.; Winarni, T. I. Physiological Effect of Deep Pressure in Reducing Anxiety of Children with ASD during Traveling: A Public Transportation Setting. Bioengineering 2022, 9, 157. https://doi.org/10.3390/bioengineering9040157
6. To enhance the understandability of the section on materials and methods easier for them to understand rather than just depending on the main text as it exists at the moment, the authors could add additional illustrations in the form of figures that explain the workflow of the present study.
7. What is the baseline of patient selection? Is there any protocol, standard, or basis that has been followed? It is unclear since the patient is very heterogeneous with a small number. The resonance involved impacts the present result makes this study flaws.
8. It is necessary to provide more information on the manufacturer, country, and specifications of the tools.
9. The revised manuscript after peer review must provide detailed information on the error and tolerance of the experimental equipment utilized in this study. Due to the disparate outcomes of other researchers' subsequent studies, it would make for a valuable discussion.
10. Outcomes must be compared to similar past research.
11. The discussion in present article is extremely poor in quality as overall. The authors must elaborate on their arguments and provide a thorough justification. Don't just state the results and give a quick explanation.
12. Please include the limitation of the present study, it is missing.
13. The reference needs to be enriched from the literature published five years back.
14. The manuscript needs to be proofread by the authors since it has grammatical and language issues.
15. It is suggested to the authors for providing graphical abstract in the system after revision.
Author Response
Response to Reviewer 1 Comments
Dear Reviewer 1:
Thanks for your kind letter about the Manuscript Number: 2067797
We have revised the manuscript following your valuable comments and carefully modified the manuscript to minimize typographical and grammatical errors.
We would like to thank the editor for your kind help and express our gratitude to the reviewers for their valuable comments. Those comments are very helpful for revising the paper, as well as the important guiding significance to our following research.
Here below are our point-to-point responses on revision according to the reviewer’s comments. For your convenience, the reviewers’ comments are displayed in blue in the responses. The main modification in the revised paper is displayed in red color.
We appreciate the editor and the reviewers, thank you again for your help. We are looking forward to getting more guidance from you.
Yours sincerely,
Wei Cheng et al.
Responses to Reviewer #1
Concern # 1: As your abstract's final sentence, include a "take-home" message.
Author response: Thank you for your great comments. We have rewritten the abstract in lines 28-31 in the new manuscript as follows:
The Frontal_Sup_L GMV played a partial mediating role in the relationship between overweight/obesity and SC, accounting for 36.6% of total effect values. These findings indicate a significant positive correlation between overweight/obesity and SC; GMV in the left dorsolateral superior frontal gyrus plays a mediating role in the relationship between overweight/obesity and SC. The study may provide new evidence toward comprehensively revealing that overweight/obesity and SC relationship.
Concern # 2: Rearrange keywords alphabetically.
Author response:
According to your great comments, we have rearranged keywords alphabetically as follows:
Keywords: children with autism spectrum disorder; gray matter volume; mediating effect; overweight /obesity; social communication
Concern # 3: Nothing truly unique in its current state. Because of the lack of novel, the current study looks to be a replication or modified study. The authors must describe their novel in detail. This work should be rejected owing to a major issue.
Author response:
Thank you for your valuable comments, we have revised the introduction in response to your suggestions, previous studies have only provided some indirect evidence for social impairment in children with ASD, and no studies have directly proved it, so this paper has two main innovations: 1. Exploration of correlation between overweight/obesity and social impairment in children with ASD. 2. The neural mechanism between overweight/obesity and social impairment in children with ASD have not been revealed yet.
Concern # 4: Previous study related needs to explain in the introduction section consisting of their work, their novelty, and their limitations to show the research gaps that intend to be filled in the present study.
Author response: Thank you for your valuable comments, and in response to your suggestions, we have revised the introduction line 44-57 and line 97-105 in the new manuscript as follows:
According to previous studies, children with ASD were found to have a significant social impairment, which can lead to problems in their social skills, communication, and interaction with others. While previous studies initially revealed that overweight/obesity may be a potential factor influencing social impairment, we inferred that the presence of social impairment in ASD may be influenced by overweight/obesity, but previous studies did not directly reveal the relationship between the two, and the neural mechanisms between the two have not been revealed, so this paper makes inferences based on previous studies. In this paper, we propose two research hypotheses: 1. there is a correlation between overweight/obesity and social impairment in children with ASD. 2. there is a mediating role of GMV in this relationship between overweight/obesity and SC.
Concern # 5: The authors need to explain some treatment in children with autism, one of the with sensory hug machine. It is a vital topic that authors must provide in the introduction and/or discussion section. Additionally, the MDPI's suggested reverence should be taken to substantiate this explanation as follows: Afif, I. Y.; Manik, A. R.; Munthe, K.; Maula, M. I.; Ammarullah, M. I.; Jamari, J.; Winarni, T. I. Physiological Effect of Deep Pressure in Reducing Anxiety of Children with ASD during Traveling: A Public Transportation Setting. Bioengineering 2022, 9, 157. https://doi.org/10.3390/bioengineering9040157.
Author response: Thank you for your valuable suggestions. We have added the way to improve treatment options for children with ASD to improve socialization in the discussion section, specifically in lines 296-297 as follows:
For example, with the help of machine therapy, highly sensitive children with ASD can be improving their social interaction and reduce sedentary time [48]
Concern # 6: To enhance the understandability of the section on materials and methods easier for them to understand rather than just depending on the main text as it exists at the moment, the authors could add additional illustrations in the form of figures that explain the workflow of the present study.
Author response: Thank you for your valuable advice. We have added an experimental flow chart in the new manuscript as follows:
In this study, we recruited 147 children aged 3 to 12 years who were diagnosed with ASD according to DSM-5 criteria from the Chuying Child Development Center, Starssailor Educational Institution, and Maternal and Child Health Hospital(Yangzhou, China). Finally, only 101 participants completed the study(Figure 1).
Inclusion criteria: (1) Han ethnicity; (2) Age 3 to 12 years old; (3) Use of the Subject Checklist(V2.0) of the Brain Imaging Center of Affiliated Hospital of Yangzhou University to check if the conditions for sMRI scanning are satisfied, including the absence of metal implants(such as metallic dentures), electronic, magnetic, or mechanical devices (such as pacemakers).
Exclusion criteria: (1) Head injury; (2) Nervous system disorders like phenylketonuria, epilepsy, tics, mental illness, etc; (3) Impaired hearing and vision; (4) Use of drugs that affect the central nervous system.
After further screening and exclusion based on experimental criteria, thirty-five children were excluded due to the above-mentioned reasons or declined to participate. And it was found that after the collection of behavioral and brain data, head movement in 7 children’s sMRI data was greater than 3mm, and 4 children had not undergone an sMRI scan. In total, 101 children with ASD were included in the study, of which 89 were male(88.12%) and 12 female(11.88%). The parents or guardians of all children who participated in this trial have signed an informed consent form according to the provision of the Declaration of Helsinki. The study was approved by the Ethics and Human Protection Committee of the Affiliated Hospital of Yangzhou University and was registered(ChiCTR190002497) in the Chinese Clinical Trials Registry
Figure 1. Experimental flow chart
Concern # 7: What is the baseline of patient selection? Is there any protocol, standard, or basis that has been followed? It is unclear since the patient is very heterogeneous with a small number. The resonance involved impacts the present result makes this study flaws.
Author response: Thank you for your valuable comments. In response to your questions, we have made a relevant experimental flow chart in the methods section about the benchmark for patient selection in this experiment is children diagnosed with ASD by DSM-5 criteria, aged 3-12 years. The inclusion and exclusion criteria of this study were also described, and all parents of the subjects signed the informed consent form. The study was also approved by the Ethics and Human Protection Committee of the Affiliated Hospital of Yangzhou University and registered by the Chinese Clinical Trials Registry. The details are described in more detail in Manuscript Methods. Specifically, in lines 106-129 of the new manuscript.
- Methods
2.1. Research Subjects
In this study, we recruited 147 children aged 3 to 12 years who were diagnosed with ASD according to DSM-5 criteria from the Chuying Child Development Center, Starssailor Educational Institution, and Maternal and Child Health Hospital(Yangzhou, China). Finally, only 101 participants completed the study (Figure 1).
Inclusion criteria: (1) Han ethnicity; (2) Age 3 to 12 years old; (3) Use of the Subject Checklist(V2.0) of the Brain Imaging Center of Affiliated Hospital of Yangzhou University to check if the conditions for sMRI scanning are satisfied, including the absence of metal implants (such as metallic dentures), electronic, magnetic, or mechanical devices (such as pacemakers).
Exclusion criteria: (1) Head injury; (2) Nervous system disorders like phenylketonuria, epilepsy, tics, mental illness, etc; (3) Impaired hearing and vision; (4) Use of drugs that affect the central nervous system.
After further screening and exclusion based on experimental criteria, thirty-five children were excluded due to the above-mentioned reasons or declined to participate. And it was found that after the collection of behavioral and brain data, head movement in 7 children’s sMRI data was greater than 3mm, and 4 children had not undergone an sMRI scan. In total, 101 children with ASD were included in the study, of which 89 were male (88.12%) and 12 female (11.88%). The parents or guardians of all children who participated in this trial have signed an informed consent form according to the provision of the Declaration of Helsinki. The study was approved by the Ethics and Human Protection Committee of the Affiliated Hospital of Yangzhou University and was registered(ChiCTR190002497) in the Chinese Clinical Trials Registry.
Concern # 8: It is necessary to provide more information on the manufacturer, country, and specifications of the tools.
Author response: Thank you for your valuable advice. We have made the following changes to the details of the test tools in the new manuscript:
All participants' height and weight were measured using the Meilen Smart Height and Weight Scale (Model: MSG005-H), Manufacturer: Shenzhen Mobil Electronics Wired Company, the error range of the experimental instrument is 0.5cm and 0.1kg, which can automatically save data and ensure the accuracy of the test data. Two physical education disciplined staff were authorized to ensure the consistency of the experimental testers during the test, and all subjects were requested to wear light clothing and undress their shoes and socks during the measurement.
Concern # 9: The revised manuscript after peer review must provide detailed information on the error and tolerance of the experimental equipment utilized in this study. Due to the disparate outcomes of other researchers' subsequent studies, it would make for a valuable discussion.
Author response: Thank you for your excellent comments, we have included the error and tolerance information related to this experimental device next to the test instrument and the questionnaire has the corresponding reliability.
All participants' height and weight were measured using the Meilen Smart Height and Weight Scale (Model: MSG005-H), Manufacturer: Shenzhen Mobil Electronics Wired Company, the error range of the experimental instrument is 0.5cm and 0.1kg,
Concern # 10: Outcomes must be compared to similar past research.
Author response: Thank you for your valuable comments. We have revised the results section based on your comments in lines 209-210 and discussed the previous findings in lines 257-341.
Concern # 11: The discussion in present article is extremely poor in quality as overall. The authors must elaborate on their arguments and provide a thorough justification. Don't just state the results and give a quick explanation.
Author response: Thank you for your valuable comments, this article focuses on the discussion section in the new manuscript has been reformulated, as seen in lines 256-344
Concern # 12: Please include the limitation of the present study, it is missing.
Author response: Thank you for your valuable comments. We have provided three limitations and a prospective depiction of this study in the new manuscript as follows:
4.3. Research limitation and outlook
This study expands on previous studies that primarily preliminarily confirmed the association between overweight/obesity and SC in ASD children, and reveals that GMV may be one of the neural mechanisms of the association between overweight/obesity and SC. However, the present study also has some limitations. First, the sample size of overweight/obese children with ASD was relatively small. The study population included children with ASD, which involved difficulties in recruiting participants. Second, this paper is mainly a preliminary exploration of the whole-brain neural mechanisms between overweight/obesity and social impairment, without analysis of brain regions of interest, and in the future, analysis of interest in both can be considered. Third, this study is a cross-sectional study. Longitudinal interventions can be used in future studies to further confirm the neural mechanisms of this study. The factors affecting SC are complex and diverse, and subsequent studies can consider the effects of other factors such as molecular and cellular levels.
Concern # 13: The reference needs to be enriched from the literature published five years back.
Author response: Thanks for your valuable comments, we have updated the references for the last five years based on your comments as follows:
- Montiel-Nava, C.; Tregnago, M.; Marshall, J.; Sohl, K.; Curran, A.B.; Mahurin, M.; Warne-Griggs, M.; WHO CST Team; Dixon, P. Implementing the WHO Caregivers Skills Training Program with Caregivers of Autistic Children via Telehealth in Rural Communi-ties. Front. Psychiatry 2022, 13, 909947, doi:10.3389/fpsyt.2022.909947.
- Healy, S.; Haegele, J.A.; Grenier, M.; Garcia, J.M. Physical Activity, Screen-Time Behavior, and Obesity Among 13-Year Olds in Ireland with and without Autism Spectrum Disorder. J Autism Dev Disord 2017, 47, 49–57, doi:10.1007/s10803-016-2920-4.
- Ridgeway, S.O. Body Size and Well-Being in Adolescents: The Roles of Bullying Victimization and Body Image. Sociological Perspectives 2022, 073112142211394, doi:10.1177/07311214221139439.
- Margolis, G.; Elbaz-Greener, G.; Ruskin, J.N.; Roguin, A.; Amir, O.; Rozen, G. The Impact of Obesity on Sudden Cardiac Death Risk. Curr Cardiol Rep 2022, 24, 497–504, doi:10.1007/s11886-022-01671-y.
- Noonan, R.J. The Influence of Adolescent Sports Participation on Body Mass Index Tracking and the Association between Body Mass Index and Self-Esteem over Three years. IJERPH 2022, 19, 15579, doi:10.3390/ijerph192315579.
- Kahathuduwa, C.N.; West, B.D.; Blume, J.; Dharavath, N.; Moustaid‐Moussa, N.; Mastergeorge, A. The Risk of Overweight and Obesity in Children with Autism Spectrum Disorders: A Systematic Review and Meta‐analysis. Obesity Reviews 2019, 20, 1667–1679, doi:10.1111/obr.12933.
- Yang, R.B.; Liu, X.J.; Han, Y.; Qu, Z.Y.; Li, Y.; Xiong, W.J.; Zhang, X.Study on physical development and nutritional status of children with autism spectrum disorders aged 3-7 years in Tianjin. Chinese Journal of Child Health Care 2021, 29, 1058-1062+1081.
- Buro, A.W.; Salinas-Miranda, A.; Marshall, J.; Gray, H.L.; Kirby, R.S. Correlates of Obesity in Adolescents with and without Autism Spectrum Disorder: The 2017–2018 National Survey of Children’s Health. Disability and Health Journal 2022, 15, 101221, doi:10.1016/j.dhjo.2021.101221.
- Li, C. R.; Li, Q.; Zhang, Y. C.; Li,X.; Yu, D.; Shi, Y.X.;Fan, L.L.;Wu, L.J. Association of the obesity major gene SH2B1 with cogni-tive and social functions of autism spectrum disorders. Chin J Child Care, Sept 2021, 29(9): 960-964.
- Lotze, M.; Domin, M.; Schmidt, C.O.; Hosten, N.; Grabe, H.J.; Neumann, N. Income Is Associated with Hippocampal/Amygdala and Education with Cingulate Cortex Grey Matter Volume. Sci Rep 2020, 10, 18786, doi:10.1038/s41598-020-75809-9.
- Ecker, C.; Ginestet, C.; Feng, Y.; Johnston, P.; Lombardo, M.V.; Lai, M.-C.; Suckling, J.; Palaniyappan, L.; Daly, E.; Murphy, C.M.; et al. Brain Surface Anatomy in Adults With Autism: The Relationship Between Surface Area, Cortical Thickness, and Autistic Symptoms. JAMA Psychiatry 2013, 70, 59, doi:10.1001/jamapsychiatry.2013.265.
- Pham, C.; Symeonides, C.; O’Hely, M.; Sly, P.D.; Knibbs, L.D.; Thomson, S.; Vuillermin, P.; Saffery, R.; Ponsonby, A.-L.; the Barwon Infant Study Investigator Group Early Life Environmental Factors Associated with Autism Spectrum Disorder Symptoms in Children at Age 2 Years: A Birth Cohort Study. Autism 2022, 26, 1864–1881, doi:10.1177/13623613211068223.
- Rong, P.; Fu, Q.; Zhang, X.; Liu, H.; Zhao, S.; Song, X.; Gao, P.; Ma, R. A Bibliometrics Analysis and Visualization of Autism Spectrum Disorder. Front Psychiatry 2022, 13, 884600, doi:10.3389/fpsyt.2022.884600.
- Camasio, A.; Panzeri, E.; Mancuso, L.; Costa, T.; Manuello, J.; Ferraro, M.; Duca, S.; Cauda, F.; Liloia, D. Linking Neuroanatom-ical Abnormalities in Autism Spectrum Disorder with Gene Expression of Candidate ASD Genes: A Meta-Analytic and Network-Oriented Approach. PLoS ONE 2022, 17, e0277466, doi:10.1371/journal.pone.0277466.
- Zhao, X.; Zhu, S.; Cao, Y.; Cheng, P.; Lin, Y.; Sun, Z.; Jiang, W.; Du, Y. Abnormalities of Gray Matter Volume and Its Correlation with Clinical Symptoms in Adolescents with High-Functioning Autism Spectrum Disorder. NDT 2022, Volume 18, 717–730, doi:10.2147/NDT.S349247.
- Arunachalam Chandran, V.; Pliatsikas, C.; Neufeld, J.; O’Connell, G.; Haffey, A.; DeLuca, V.; Chakrabarti, B. Brain Structural Correlates of Autistic Traits across the Diagnostic Divide: A Grey Matter and White Matter Microstructure Study. NeuroImage: Clinical 2021, 32, 102897, doi:10.1016/j.nicl.2021.102897.
- Pereira, A.M.; Campos, B.M.; Coan, A.C.; Pegoraro, L.F.; de Rezende, T.J.R.; Obeso, I.; Dalgalarrondo, P.; da Costa, J.C.; Dreher, J.-C.; Cendes, F. Differences in Cortical Structure and Functional MRI Connectivity in High Functioning Autism. Front. Neurol. 2018, 9, 539, doi:10.3389/fneur.2018.00539.
Concern # 14: The manuscript needs to be proofread by the authors since it has grammatical and language issues.
Author response: Thank you for your valuable advice. We apologize for the poor language of our manuscript. We worked on the manuscript for a long time. We have now worked on both language and readability, and have also involved native English speakers for language corrections. We hope that the language level has been substantially improved.
Concern # 15: It is suggested to the authors for providing graphical abstract in the system after revision.
Author response: Thank you for your valuable advice. According to your comments, we have made a graphical abstract as follows:
Dear Reviewer, we really appreciate and admire your valuable help. Your responsible and careful comments are not only conducive to the improvement of this work, but also conducive to our following research. We must lucky enough to meet a reviewer like you, especially in the virus outbreak period.
We have been tried our best to revise and polish, and we really hope you could approve our efforts for this manuscript. We are looking forward to getting more guidance and help from you.
Sincerely,
Wei Cheng, et al.

Reviewer 2 Report
Comments and Suggestions for Authors
Introduction
The concept of the paper is interesting and may spark the interest of the authors. I believe my suggestions are constructive and help authors bring the manuscript to the next level after fixing them. May major comments are pertaining to the normalization of BMI as reference intervals are age-specific, hence requiring reanalysis of data.
Major comments
1. The methods section should include the protocol for anthropometric measurements.
How the height was measured(instrument)? How the weight was measured(instrument)? Who measured it? How was obesity/overweight determined? Have you used any “Han population-based” growth charts? If not, what growth chats have you used in determining obesity/overweight?
2. BMI is age-dependent, therefore mean+/-SD cannot be used to determine the subject with overweight obesity. However, the title itself states “overweight/obesity”. To justify the title the following information should be there.
a) Number and Percentage of overweight, obese, normal, and underweight, children respectively.
b) Indicate the growth chart that you used. If you don’t have charts for your own population, use international charts but state this as a limitation.
c) For the same reason stated above, run the Pearson correlation to normalized BMI (BMI percentile). Then only readers can meaningfully interpret the results.
There are online tools that can be used to derive data from charts.
d) When you analyze/summarize/discuss the statistical data use categorical variables (overweight/obesity with a text like T-test or non-parametric equivalent. With BMI percentile, Pearson correlation. Because overweight/obesity is a categorical variable, therefore Pearson correlation is not applicable.
https://www.cdc.gov/healthyweight/bmi/result.html?&method=english&gender=f&age_y=4&age_m=4&hinches=40&twp=33
Minor comments
Line 38; “Children with ASD with SC impairment” implies that there can be “Children ASD without SC impairment”.
Suggest paraphrasing “Children with ASD exhibit SC impairments in terms of deficits in social-emotional reciprocity, nonverbal communicative behaviors used for social interaction, and developing, maintaining, and understanding relationships.
Line 42-43; Not so clear. Is this
Furthermore, SC impairments “lead to” communication difficulties among peers, bullying, and depression in children with ASD
Methods section
Line 107; “children aged 3 to 12 years satisfying DSM-5 diagnostic criteria for ASD. “
Line 116; This exclusion criterion is not so clear “short-term CNS medicinal products”, clarify with an example. Does that mean being on long-term medicinal products is not an exclusion criterion?
Author Response
Response to Reviewer 2 Comments
Dear Reviewer 2,
Thanks for your kind letter about Manuscript Number: 2067797
We have studied the comments carefully and have made correction accordingly.
We would like to thank the editor for your kindly help and express our gratitude to the reviewers for their valuable comments. Those comments are very help for revising the paper, as well as the important guiding significance to our following research.
Here below is our point to point responds on revision according to the reviewer’s comments. For your convenience, the reviewers’ comments are displayed as blue in the responds. The main modification in the revised paper is displayed in red color.
We really appreciate the editor and the reviewers, thank you again for your help. We are looking forward to getting more guidance from you.
Yours sincerely,
Wei Cheng et al.
Reponses to Reviewer #2
General Comments: The concept of the paper is interesting and may spark the interest of the authors. I believe my suggestions are constructive and help authors bring the manuscript to the next level after fixing them. May major comments are pertaining to the normalization of BMI as reference intervals are age-specific, hence requiring reanalysis of data.
Author response: Thank you very much for your kind and positive comments, which motivate us to further revise and enhance the quality of this review.
Concern # 1: The methods section should include the protocol for anthropometric measurements.
How the height was measured(instrument)? How the weight was measured(instrument)? Who measured it? How was obesity/overweight determined? Have you used any “Han population-based” growth charts? If not, what growth chats have you used in determining obesity/overweight?
Author response: Thank you for your excellent comments. We have rewritten method in the new draft lines 133-142 as follows:
All participants' height and weight were measured using the Meilen Smart Height and Weight Scale (Model: MSG005-H), Manufacturer: Shenzhen Mobil Electronics Wired Company, the error range of the experimental instrument is 0.5cm and 0.1kg, which can automatically save data and ensure the accuracy of the test data. Two physical education disciplined staff were authorized to ensure the consistency of the experimental testers during the test, and all subjects were requested to wear light clothing and undress their shoes and socks during the measurement. BMI was calculated using the traditional calculation method, BMI=(weight (kg))/(height (m))2. The over-weight/obesity scoring criteria thresholds were based on the 2010 Chinese 2-18 years BMI values of children and adolescents [36].
Concern # 2: BMI is age-dependent, therefore mean+/-SD cannot be used to determine the subject with overweight obesity. However, the title itself states “overweight/obesity”. To justify the title the following information should be there.
- Number and Percentage of overweight, obese, normal, and underweight, children respectively.
Author response: Thank you for your excellent comments, we have added new content to lines 142-143 at the method, as follows.
Among the participants, 13 children were overweight (12 boys/1 girls) 12.87%, and 20 children were obese (18 boys/2 girls) 19.80%.
- b) Indicate the growth chart that you used. If you don’t have charts for your own population, use international charts but state this as a limitation.
Author response: Thank you for your comment, we have classified overweight and obese according to the following table:
(Table 1) BMI (kg/m2) cut-off point assessment criteria for children and adolescents aged 2-18 years in China
|
Year |
3 |
3.5 |
4 |
4.5 |
5 |
5.5 |
6 |
6.5 |
7 |
7.5 |
8 |
8.5 |
9 |
9.5 |
10 |
10.5 |
11 |
11.5 |
12 |
|
|
boy |
overweight |
16.8 |
16.6 |
16.5 |
16.4 |
16.5 |
16.6 |
16.8 |
17 |
17.2 |
17.5 |
17.8 |
18.2 |
18.5 |
18.9 |
19.3 |
19.7 |
20.1 |
20.4 |
20.8 |
|
Obesity |
18.1 |
17.9 |
17.8 |
17.8 |
17.9 |
18.1 |
18.4 |
18.4 |
19.2 |
19.6 |
20.1 |
20.6 |
21.1 |
21.7 |
22.2 |
22.7 |
23.2 |
23.7 |
24.2 |
|
|
girl |
overweight |
16.9 |
16.8 |
16.7 |
16.6 |
16.6 |
16.7 |
16.7 |
16.8 |
16.9 |
17.1 |
17.3 |
17.6 |
17.9 |
18.3 |
18.7 |
19.1 |
19.6 |
20.1 |
20.5 |
|
Obesity |
18.3 |
18.2 |
18.1 |
18.1 |
18.2 |
18.3 |
18.4 |
18.6 |
18.8 |
19.1 |
19.5 |
19.9 |
20.4 |
20.9 |
21.5 |
22.1 |
22.7 |
23.3 |
23.9 |
|
- For the same reason stated above, run the Pearson correlation to normalized BMI (BMI percentile). Then only readers can meaningfully interpret the results. There are online tools that can be used to derive data from charts.
Author response: Thank you for your valuable comments. In this study, we have corresponding scoring criteria at each age stage, which are also standardized for BMI, as shown in the table above.
d)When you analyze/summarize/discuss the statistical data use categorical variables (overweight/obesity with a text like T-test or non-parametric equivalent. With BMI percentile, Pearson correlation. Because overweight/obesity is a categorical variable, therefore Pearson correlation is not applicable.
Author response: Thank you for your excellent comments, the BMI value was used for the analysis. BMI is a continuous variable, and this paper was conducted on all subjects, rather than categorical variables for statistics. The sample size of this study is small, and the sample size will be expanded in the follow-up study to continue the exploration.
Concern # 3: Line 38; “Children with ASD with SC impairment” implies that there can be “Children with ASD without SC impairment”. Suggest paraphrasing “Children with ASD exhibit SC impairments in terms of deficits in social-emotional reciprocity, nonverbal communicative behaviors used for social interaction, and developing, maintaining, and understanding relationships.
Author response: Thank you for the valuable suggestions. Your suggestions have been adopted and the 38-42 lines have been revised in the new manuscript, as follows:
Autism spectrum disorder (ASD) is a lifelong, severe psychoneurotic-developmental disorder occurring in early childhood that limits or impairs daily functioning [1]. One of the core symptoms of ASD is social communication (SC) impairment. Children with ASD generally show SC impairments in understanding, maintaining, and developing social relationships [2]. It severely hinders them from making friends and leads to a consequent decline in physical activity and an increase in sedentary time [3]
Concern # 4: Lines 42-43; Not so clear. Is this
Furthermore, SC impairments “lead to” communication difficulties among peers, bullying, and depression in children with ASD.
Author response: Thank you for your valuable suggestions, lines 44-46 of the draft are not very clear, now the new manuscript is revised as follows:
Notably, prolonged sedentary time can induce anxiety or obesity, and lower the quality of life. Also, negatively impacted the SC, obese children are vulnerable to bullying from peers [4,5]. 1,490 children aged 7-12 were surveyed on their health condition, and it was found that obese children suffer from severe depression [6]. The investigation of the dietary habits of 54 children with ASD presented a 42.6% obesity rate among children with ASD and discovered a positive correlation between obesity and symptom severity [7].
Concern # 5: Line 107; “children aged 3 to 12 years satisfying DSM-5 diagnostic criteria for ASD.
Author response: Thank you for your valuable suggestions, in this paper the intended description was to recruit children diagnosed with ASD according to DSM-5 criteria, now in the new manuscript the change to line 108 is as follows:
In this study, we recruited 147 children aged 3 to 12 years who were diagnosed with ASD according to DSM-5 criteria from the Chuying Child Development Center, Starssailor Educational Institution, and Maternal and Child Health Hospital (Yangzhou, China). Finally, only 101 participants completed the study (Figure 1).
Concern # 6: This exclusion criterion is not so clear “short-term CNS medicinal products”, clarify with an example. Does that mean being on long-term medicinal products is not an exclusion criterion?
Author response: Thank you for your valuable suggestions. For line 116, on the exclusion criteria for short-term central nervous system drugs on the expression of the error, mainly to express the exclusion criteria is taking drugs that affect the central nervous system which has been revised in line 116 as follows.
Exclusion criteria: (1) head injury; (2) nervous system disorders like phenylketonuria, epilepsy, tics, mental illness, etc; (3) impaired hearing and vision; (4) Use of drugs that affect the central nervous system.
Dear Reviewer, we really appreciate and admire your valuable help. Your responsible and careful comments are not only conducive to the improvement of this work, but also conducive to our following research. We must lucky enough to meet a reviewer like you, especially in the virus outbreak period.
We have been tried our best to revise and polish, and we really hope you could approve our efforts for this manuscript. We are looking forward to getting more guidance and help from you.
Sincerely,
Wei Cheng, et al.

Reviewer 3 Report
Comments and Suggestions for Authors
In this study, the authors investigate the association between obesity, social difficulties (SC) and grey matter volume (GMV) in children on the autism spectrum. Although the topic might be of interest for medical research on autism, I have some major concerns that are worth the authors' careful consideration.
1. Abstract and Introduction
a. The abstract could be more informative: in addition to reporting the results, it would be useful to give a brief indication of their interpretation.
b. The introduction places the study within the medical model of autism. Although this is congruent with the objectives of the study, I would point out that this could raise criticism from neurodiversity movements that advocate a social model of autism, and the use of less medicalising language (https://www.liebertpub.com/doi/full/10.1089/aut.2020.0014). I simply invite the authors to take this debate into consideration in case they are not aware of it.
c. I am not an expert on obesity nor GMV, so I am not confident in assessing whether the theoretical background has been presented completely and accurately, and whether there is indeed a gap in the literature that justifies this study. However, the review of previous literature seems scanty to me. For instance, the authors state that: “Numerous studies have shown a close association between overweight/obesity and SC in children with ASD” However, there are no specific references to these numerous previous studies.
d. I would recommend the authors to clarify the explication of the study's hypotheses. What results did the researchers expect? What contribution did they aim to make to the scientific community and what specific benefit to autism research?
2. Methods:
a. There is no information on how the sample size was chosen. In addition, a very small percentage of the more than 100 children tested were overweight or obese. I wonder if the sample is suitable for the purposes of the study. Even more importantly, one of the major limitations of the study is the absence of a neurotypical or otherwise non-autistic comparison group. This means that the results obtained, assuming they are informative, cannot be attributed exclusively to the autistic population.
b. Pearson correlations have been run. However, this correlation type can only be used when all variables are normally distributed. No information on the variable distributions can be found in the manuscript.
c. The authors state: “The study aimed to analyze the mediating role of gray matter volume 100 in the relationship between overweight/obesity and SC and to reveal the relationship between overweight/obesity and SC in children with ASD.” And then in the results: “Given the length of the correlation analysis of GMVs in 116 brain regions, and the other brain regions the following was not placed in this paper (Table 1).” It seems to me there is a bad fishing practice behind this decision to rely solely on statistical significance to include or exclude results from a scientific paper. I wonder why the authors didn’t focus on brain areas of interest (based on the literature), conduct correlation analyses on them and fully report the results (considering adding Supplementary Materials to the main manuscript, if necessary). In the same vein, I would invite authors to share raw data on a public repository, supporting transparency, reproducibility, and open science. The current data availability statement is questionable: "The data presented in this study are available on request from the corresponding author. The data are not publicly available due to the privacy of participants.". Many clinical studies worldwide share their data upon anonimisation of participants' information.
d. Table 1 and Figure 1 should report units of measurement of each study variable.
e. When reporting p-values, there is no need to include “<0.05”. For instance, “p =0.01<0.05” should rather be “p =0.01”
3. Discussion and conclusions
a. Line 323: “However, our study results were inconsistent with previous findings, which may be attributed to the heterogeneity of ASD and the age difference of the study subjects.” Further explanation of the similarity and differences between the present work and other means in the literature is needed. such explanations cannot be based exclusively on the heterogeneity of the autistic population. or, if the authors believe that each scientific work on autism obtains results exclusively relevant to the sample involved, this implies that no previous results (nor those presented in this work) can be generalised to the autistic population as a whole.
b. “The presence of overweight/obesity in children with ASD may lead to a decrease in GMV in Frontal_Sup_L.” Did the authors test the other direction of this association? How can they rule out the possibility that a reduced GMV is what leads to obesity? The same goes for the association between obesity and social difficulties. How do we know that (as the authors conclude) “overweight/obese children with ASD may be at greater risk of SC impairment” and not vice versa (i.e., social difficulties come with increased risk of obesity)?
Since a correlation does not imply causation, how can the present correlation analysis be used to demonstrate directionality of any of the associations of interest?
Author Response
Response to Reviewer 3 Comments
Dear Reviewer 3,
Thanks for your kind letter about Manuscript Number: 2067797
We have studied the comments carefully and have made correction accordingly.
We would like to thank the editor for your kindly help and express our gratitude to the reviewers for their valuable comments. Those comments are very help for revising the paper, as well as the important guiding significance to our following research.
Here below is our point to point responds on revision according to the reviewer’s comments. For your convenience, the reviewers’ comments are displayed as blue in the responds. The main modification in the revised paper is displayed in red color.
We really appreciate the editor and the reviewers, thank you again for your help. We are looking forward to getting more guidance from you.
Yours sincerely,
Wei Cheng et al.
Reponses to Reviewer #3
General Comments: In this study, the authors investigate the association between obesity, social difficulties (SC) and grey matter volume (GMV) in children on the autism spectrum. Although the topic might be of interest for medical research on autism, I have some major concerns that are worth the authors' careful consideration.
Author response: Thank you very much for your kind and positive comments, which motivate us to further revise and enhance the quality of this article.
Concern # 1: Abstract and Introduction.
- The abstract could be more informative: in addition to reporting the results, it would be useful to give a brief indication of their interpretation.
Author response: Thank you for your valuable comments. We have rewritten the abstract 28-34 lines in the new manuscript as follows:
The Frontal_Sup_L GMV played a partial mediating role in the relationship between overweight/obesity and SC, accounting for 36.6% of total effect values. These findings indicate a significant positive correlation between overweight/obesity and SC; GMV in the left dorsolateral superior frontal gyrus plays a mediating role in the relationship between overweight/obesity and SC. The study may provide new evidence toward comprehensively revealing the overweight/obesity and SC relationship.
- The introduction places the study within the medical model of autism. Although this is congruent with the objectives of the study, I would point out that this could raise criticism from neurodiversity movements that advocate a social model of autism, and the use of less medicalising language (https://www.liebertpub.com/doi/full/10.1089/aut.2020.0014). I simply invite the authors to take this debate into consideration in case they are not aware of it.
Author response: Thank you for your valuable comments. Your suggestions have enriched my understanding of autism. And I will continue to improve my professional language in my future research.
- I am not an expert on obesity nor GMV, so I am not confident in assessing whether the theoretical background has been presented completely and accurately, and whether there is indeed a gap in the literature that justifies this study. However, the review of previous literature seems scanty to me. For instance, the authors state that: “Numerous studies have shown a close association between overweight/obesity and SC in children with ASD” However, there are no specific references to these numerous previous studies.
Author response: Thank you for your valuable comments, through previous studies, we found that children with ASD have a severe social impairment, and overweight/obesity affects social impairment, but there is no direct evidence to confirm the relationship between overweight/obesity and social impairment. This paper also presents hypotheses based on previous studies to confirm the direct relationship between overweight/obesity and social impairment in children with ASD. It is in the preliminary exploration stage, and a large part of the introduction of this article has been revised in line 38-105.
Autism spectrum disorder(ASD) is a lifelong, severe psychoneurotic-developmental disorder occurring in early childhood that limits or impairs daily func-tioning[1]. One of the core symptoms of ASD is social communication(SC) impairment. Children with ASD generally show SC impairments in understanding, maintaining, and developing social relationships [2]. It severely hinders them from making friends and leads to a consequent decline in physical activity and an increase in sedentary time [3]. Notably, prolonged sedentary time can induce anxiety or obesity, and lower the quality of life. Also, negatively impacted the SC, obese children are vulnerable to bullying from peers [4,5]. 1,490 children aged 7-12 were surveyed on their health con-dition, and it was found that obese children suffer from severe depression [6]. The in-vestigation of the dietary habits of 54 children with ASD presented a 42.6% obesity rate among children with ASD and discovered a positive correlation between obesity and symptom severity [7]. Through the study of children with ASD aged 3-7 years, Yang et al. found that the severity of symptoms can be accompanied by severe diet be-haviors that lead to a rapid increase in individual BMI, and then abnormal sensory, cognitive, behavioral, and adaptive functioning in children with ASD [8]; Based on this, they concluded that social impairment in children with ASD may strongly associated with a rapid increase in BMI. By examining the prevalence and health correlates of overweight/obesity in children with ASD, scholars have found that overweight/obese children with ASD participate in fewer physical activities, skill development, leading to poor physical coordination, and weak communication skills [9].
Generally, many pieces of research found that overweight/obese children with ASD may also exhibit a range of abnormal responses, possibly social isolation or ag-gressive behavior. Notably, a study in 2021 on obesity genes and social functioning in ASD confirmed a link between obesity genes and social functioning [10]. However, previous studies have only suggested that overweight/obesity may be a potential factor influencing social impairment, and the relationship between them has not been con-firmed. Therefore, the first hypothesis in this study is proposed: there may be a significant correlation between overweight/obesity and SC in children with ASD.
Up to now, the neural mechanisms affecting SC in children with ASD have be-come a hot research spot. Social communication in children with ASD can be affected by different mechanisms, which are mostly related to neural mechanisms [11–13]. Typ-ically, Gray matter volume(GMV) has been widely studied as a neural mechanism. The "social brain" theory of Brothers revealed that brain regions are closely related to social functioning where reduced GMV in children with ASD is the key to SC impairment in children with ASD [14]. Growing evidence demonstrates abnormal GMV in several brain regions in ASD patients, including the frontal, temporal, parietal, anterior cingu-late, insula, caudate nucleus, and lingual gyrus [15–17]. During the growth of children with ASD, lower GMV has been observed in the inferior and superior frontal gyri, me-dial and superior occipital gyri, hippocampus, and the left hemisphere of the epen-cephalon, frequently leading to SC impairment[18,19]. There is a strong association be-tween SC impairment and reduced GMV in the frontal lobe, frontal-trigeminal cerebel-lar region, and superior temporal sulcus in children with ASD [20,21]. GMV has a strong correlation with SC in children with ASD. However, whether GMV is a neural mechanism between overweight/obesity and SC in children with ASD has not been re-vealed.
Research in the field of brain science has concluded that being overweight/obesity causes poor brain health [22], such as reduced gray matter and white matter. The World Health Organization and the National Center for Health Statistics apply BMI as an indicator to define overweight/obesity. Research evidence reveals that children with ASD are at greater risk for obesity [23]. It was illustrated that obesity not only poses a threat to the growth and nutrition of children [24], but also contributes to nu-merous diseases such as hypertension [25], diabetes [26], and many other dangerous diseases[27]. A review study in 2021 found a significant negative association between overweight/obesity and GMV [28]. A 1-year follow-up of adolescents demonstrated that the increase in BMI is associated with a decrease in frontal GMV [29]. The same study explored the relationship between childhood obesity and GMV, indicating that changes in GMV in the pallidum [30,31], thalamus, cerebellum, amygdala[32], cingu-late [33], and hippocampus [34]regions are associated with obesity in children. Moreo-ver, a review found that elevated BMI in adolescents is associated with reduced GMV, mainly in prefrontal and limbic regions [35].
In summary, overweight/obesity is significantly negatively correlated with GMV. However, the relationship between overweight/obesity and GMV among children with ASD has not been conclusively established.
With the development of brain imaging technology, brain plasticity has been con-firmed. Currently, correlational researches make it a hotspot to utilize GMV as a medi-ator in the methodology. While, the relationship between overweight/obesity, GMV, and SC in children with ASD has not been fully revealed, especially whether GMV is a potential neural mechanism for overweight/obesity and SC remains unclear. Therefore, hypothesis 2 in this study is proposed: there is a mediating role of GMV in this rela-tionship between overweight/obesity and SC.
- I would recommend the authors to clarify the explication of the study's hypotheses. What results did the researchers expect? What contribution did they aim to make to the scientific community and what specific benefit to autism research?
Author response: Thank you for your excellent comments. Two research hypotheses of this study have been revised in our new manuscript, and the following are elaborated in the new manuscript:
- There is a correlation between overweight/obesity and social impairment in children with ASD;
- The neural mechanisms between overweight/obesity and social impairment in children with ASD have not been revealed.
Through previous studies, we found that obesity is some potential factor affecting social impairment. However, there is no study to confirm the direct relationship between overweight/obesity and social impairment in children with ASD, and the neural between overweight/obesity and social impairment has not been revealed. We found a significant relationship between overweight/obesity and social impairment through hypothesis 1 and 2 of this study and revealed that gray matter volume is one of the neural mechanisms between overweight/obesity and social impairment. The findings of this study may have a little public health implication, as our study suggests that overweight/obese children with ASD are vulnerable to social impairment in the future, and gray matter volume is one of the neural mechanisms between overweight/obesity and social impairment. It can also provide intervention ideas to improve social impairment in children with ASD.
Concern # 2: Methods:
- There is no information on how the sample size was chosen. In addition, a very small percentage of the more than 100 children tested were overweight or obese. I wonder if the sample is suitable for the purposes of the study. Even more importantly, one of the major limitations of the study is the absence of a neurotypical or otherwise non-autistic comparison group. This means that the results obtained, assuming they are informative, cannot be attributed exclusively to the autistic population.
Author response: Thank you for your suggestion, we have added specific sample inclusion and exclusion criteria in the methods section, and also have added the relevant experimental flow chart. In this study, we mainly found obesity as a potential factor affecting social impairment based on previous studies, but there is no direct relationship to prove the relationship between overweight/obesity and social impairment. The sample size of this study is small, and it is also suggested in the study limitations, and the sample size will be expanded in the follow-up study to continue the exploration. Our focus was on the relationship between overweight/obesity and social impairment in children with autism, rather than the differences between children with ASD and normal children for comparison; So, normal children were not selected as a control group, and we will consider adding a control group for comparison in future studies when we expand the sample.
- Pearson correlations have been run. However, this correlation type can only be used when all variables are normally distributed. No information on the variable distributions can be found in the manuscript.
Author response: Thanks for your valuable comments, we have added the information about the orthogonal distribution test in lines 191-192 of the new manuscript.
- The authors state: “The study aimed to analyze the mediating role of gray matter volume 100 in the relationship between overweight/obesity and SC and to reveal the relationship between overweight/obesity and SC in children with ASD.” And then in the results: “Given the length of the correlation analysis of GMVs in 116 brain regions, and the other brain regions the following was not placed in this paper (Table 1).” It seems to me there is a bad fishing practice behind this decision to rely solely on statistical significance to include or exclude results from a scientific paper. I wonder why the authors didn’t focus on brain areas of interest (based on the literature), conduct correlation analyses on them and fully report the results (considering adding Supplementary Materials to the main manuscript, if necessary). In the same vein, I would invite authors to share raw data on a public repository, supporting transparency, reproducibility, and open science. The current data availability statement is questionable: "The data presented in this study are available on request from the corresponding author. The data are not publicly available due to the privacy of participants.". Many clinical studies worldwide share their data upon anonimisation of participants' information.
Author response: Thank you for your valuable comments, I am very sorry to bring you such a misunderstanding, this paper is mainly to confirm the relationship between overweight/obesity and social impairment in children with ASD. In previous studies, it was found that the neural mechanism between overweight obesity and social impairment has not been revealed yet, and the relationship discovered in this article between overweight/obesity and social impairment is only a preliminary exploration. This paper is not aimed at exploration to a certain brain area, but whole-brain related exploration. It is also proposed in our limitations that the area of interest analysis of the relationship between overweight/obesity and social impairment can be performed in future studies. So, the brain regions of interest related to both are reported directly, and this sentence has been removed from the article, thank you very much for your suggestion.
We apologize for the data sharing because we have signed the agreement with family before the experiment that no information about the child would be disclosed, and we will pay attention to such issues in subsequent studies.
- Table 1 and Figure 1 should report units of measurement of each study variable.
Author response: Thank you for your valuable comments, the units of measurement in the new manuscript have been revised.
- When reporting p-values, there is no need to include “<0.05”. For instance, “p=0.01<0.05” should rather be “p=0.01”
Author response: Thank you for your valuable comments, the new manuscript has been revised at the p-value of the full text.
Concern # 3: Discussion and conclusions
- Line 323: “However, our study results were inconsistent with previous findings, which may be attributed to the heterogeneity of ASD and the age difference of the study subjects.” Further explanation of the similarity and differences between the present work and other means in the literature is needed. such explanations cannot be based exclusively on the heterogeneity of the autistic population. or, if the authors believe that each scientific work on autism obtains results exclusively relevant to the sample involved, this implies that no previous results (nor those presented in this work) can be generalised to the autistic population as a whole.
Author response: Thank you for your wonderful comment, we have modified the discussion section 304-324 lines according to your suggestion as follows:
Studies on the social brain network studies in adults with ASD revealed that reduced GMV in several brain regions, such as the right inferior occipital gyrus, left syrinx gyrus, right middle temporal gyrus, bilateral amygdala, right inferior frontal gyrus, right orbitofrontal lobe, and left dorsomedial prefrontal lobe, impacts their SC [52,53]. The result in this article is consistent with existing studies, and our study shows that GMV of the left dorsolateral superior frontal gyrus is also one of the neural mechanisms affecting SC, where numerous findings also suggest that GMV of the left dorsolateral superior frontal gyrus plays an important role in the brain. Firstly, it has been suggested that the dorsolateral superior frontal gyrus is closely related to emotion regulation, cognitive processes, and self-awareness [54], providing a higher neural basis for cognitive function. The dorsolateral superior frontal gyrus may also be crucial to SC in children with ASD. Secondly, the left dorsolateral superior frontal gyrus is an important component brain area of the default mode network (DMN), and DMN abnormalities are closely related to social interaction and verbal communication impairment in ASD [55]. In addition, the activity of the DMN is closely related to the memory, emotion, and cognitive functions of the human brain. The reduced GMV of the left dorsolateral superior frontal gyrus impairs the normal activity of the DMN [56–58], which may lead to poorer SC in children with ASD. Findings from previous studies have discovered a strong association between reduced GMV in the inferior frontal gyrus, amygdala, and thalamus and social impairment in children with ASD. Little strong association in this study was found, it probably attributed to the discrepancy of subjects (subjects in this article were younger children, but their studies were predominantly adults) [59]. Based on the analysis above, the present study also verified that GMV of the left dorsolateral superior frontal gyrus significantly affects SC in children with ASD through neural mechanisms.
- “The presence of overweight/obesity in children with ASD may lead to a decrease in GMV in Frontal_Sup_L.” Did the authors test the other direction of this association? How can they rule out the possibility that a reduced GMV is what leads to obesity? The same goes for the association between obesity and social difficulties. How do we know that (as the authors conclude) “overweight/obese children with ASD may be at greater risk of SC impairment” and not vice versa (i.e., social difficulties come with increased risk of obesity)?
Author response: Thank you for your advice, which has opened my mind further in the direction of research. This paper is to confirm the correlation between overweight/obesity and social impairment in children with ASD, and the study found that there is a positive correlation between them. The premise is to do a retrospective analysis of them. Under certain conditions, it is possible to explain the relationship in turn, but this study does not confirm the other direction of the association. In previous studies, obesity and gray matter volume are based on the results of the comparison of the two groups of experiments. In future, we will also further work on this issue in this direction.
C Since a correlation does not imply causation, how can the present correlation analysis be used to demonstrate the directionality of any of the associations of interest?
Author response: Thank you for your valuable comments. This study is only an exploratory cross-sectional study, and in future studies, we can do a longitudinal intervention study to prove our findings.
Dear Reviewer, we appreciate and admire your valuable help. Your responsible and careful comments are not only conducive to the improvement of this work but also conducive to our following research. We must lucky enough to meet a reviewer like you, especially during the virus outbreak period.
We have tried our best to revise and polish it, and we hope you could approve our efforts for this manuscript. We are looking forward to getting more guidance and help from you.
Sincerely,
Wei Cheng, et al.

Round 2
Reviewer 1 Report
Comments and Suggestions for Authors
Well done. I do not have and further comments for the manuscript at this stage.
Author Response
Dear academic editor:
Thank you for all your suggestions. We cherish them very much. Thank you.
Sincerely,
Wei Cheng
Reviewer 2 Report
Comments and Suggestions for Authors
Dear Authors,
Congratulations ! you have fixed almost all my suggestions.
I have only a few more minor suggestions you may consider;
comment 1
Authors have used BMI and Overweight/Obesity interchangeably in many statements. But BMI is a continuous variable and the state of being obese/overweight is categorical/ordinal
In your statistical analysis (Partial correlation) you have used BMI not the categorical/ordinal variable (normal/overweight/obese).
So it's actually the correlation between BMI and SC but the authors state right throughout the manuscript "Correlation between overweight/obesity and ....). even in the figure legends of graphs clearly showing it the BMI, not the category (normal/Obese/overweight). I hope this makes sense. It's true that BMI correlates with obesity but in terms of statistical principles and scientific soundness authors need to be precise about what exact variable they used in the analysis. BMI or the category of nutritional status? It's BMI isn't it?
comment 2;
1. Cite primary references (original research or a systematic review/meta-analysis directly linked to the point of discussion) but not a secondary source which discusses the finding of a primary source.
Line 85-86; for instance when discussing "Research evidence reveals that children with ASD are at greater risk for obesity".
But, you have cited a paper on "Orthopedic Problems in Overweight and Obese Children".
You could have cited "Sammels O, Karjalainen L, Dahlgren J, Wentz E. Autism Spectrum Disorder and Obesity in Children: A Systematic Review and Meta-Analysis. Obes Facts. 2022;15(3):305-320. doi: 10.1159/000523943. Epub 2022 Mar 9. PMID: 35263756; PMCID: PMC9210004."
Comment 3
And I see some references published exclusively in the Chinese language.
E.g. (24) Li,Y.; Liang,S.;Han,P.P.;Yao,X.;Li,H.X.;Sun,Y.;Jiang,X.T.;Wu,L.J.A Case-conteol Study on Boby Composition of ASD Children. Chinese Journal of School Health 2016,37(06),831-834 463 DOI:10.16835/j.cnki.1000-9817.2016.06.008
I think it is good to cite a paper published in English along with what you have already cited as the journal is international and should interest a broader audience.
Does Reference [8] provide evidence to support your statement indicated below? (This is again a Chinese paper without any English translations available online, hence couldn't read the full text). Does this paper investigate the relationship between dietary behaviour, BMI and abnormal sensory, cognitive, behavioural, and adaptive functioning? I think the study investigates primarily the nutritional status.
Line 50-52; "severity of symptoms can be accompanied by severe diet behaviors that lead to a rapid increase in individual BMI, and then abnormal sensory, cognitive, behavioural, and adaptive functioning in children with ASD [8]"
I suggest going through the references carefully to see if they support the statements as a primary source of evidence and fixing other similar issues that you might identify.
Author Response
Response to Reviewer 2 Comments
Dear academic editor:
Thank you for your letter dated January 9. We thank the reviewers for the time and effort that they have put into reviewing the previous version of the manuscript. Their suggestions have enabled us to improve our work. Based on the instructions provided in your letter, we uploaded the file of the revised manuscript. Accordingly, we have uploaded a copy of the original manuscript with all the changes highlighted by using the track changes mode in MS Word. Appended to this letter is our point-by-point response to the comments raised by the reviewers. The comments are reproduced and our responses are given directly afterward in a different color (red). We would like also to thank you for allowing us to resubmit a revised copy of the manuscript.
Sincerely,
Wei Cheng
Point 1: Authors have used BMI and Overweight/Obesity interchangeably in many statements. But BMI is a continuous variable and the state of being obese/overweight is categorical/ordinal.
In your statistical analysis (Partial correlation) you have used BMI not the categorical/ordinal variable (normal/overweight/obese).
So it's actually the correlation between BMI and SC but the authors state right throughout the manuscript "Correlation between overweight/obesity and ....). even in the figure legends of graphs clearly showing it the BMI, not the category (normal/Obese/overweight). I hope this makes sense. It's true that BMI correlates with obesity but in terms of statistical principles and scientific soundness authors need to be precise about what exact variable they used in the analysis. BMI or the category of nutritional status? It's BMI isn't it?
Response 1: Thank you very much for your valuable comments. The article examines overweight/obesity in ASD by BMI, mainly trying to explain it by high BMI. Based on previous literature, the overweight/obesity expressed by BMI was emphasized in the article.
Point 2: Cite primary references (original research or a systematic review/meta-analysis directly linked to the point of discussion) but not a secondary source which discusses the finding of a primary source.
Line 85-86; for instance when discussing "Research evidence reveals that children with ASD are at greater risk for obesity".
But, you have cited a paper on "Orthopedic Problems in Overweight and Obese Children".
You could have cited "Sammels O, Karjalainen L, Dahlgren J, Wentz E. Autism Spectrum Disorder and Obesity in Children: A Systematic Review and Meta-Analysis. Obes Facts. 2022;15(3):305-320. doi: 10.1159/000523943. Epub 2022 Mar 9. PMID: 35263756; PMCID: PMC9210004."
Response 2: Thank you for your great suggestion, we have made that change of reference in the article
Point 3: And I see some references published exclusively in the Chinese language.
E.g. (24) Li,Y.; Liang,S.;Han,P.P.;Yao,X.;Li,H.X.;Sun,Y.;Jiang,X.T.;Wu,L.J.A Case-conteol Study on Boby Composition of ASD Children. Chinese Journal of School Health 2016,37(06),831-834 463 DOI:10.16835/j.cnki.1000-9817.2016.06.008
I think it is good to cite a paper published in English along with what you have already cited as the journal is international and should interest a broader audience.
Does Reference [8] provide evidence to support your statement indicated below? (This is again a Chinese paper without any English translations available online, hence couldn't read the full text). Does this paper investigate the relationship between dietary behaviour, BMI and abnormal sensory, cognitive, behavioural, and adaptive functioning? I think the study investigates primarily the nutritional status.
Line 50-52; "severity of symptoms can be accompanied by severe diet behaviors that lead to a rapid increase in individual BMI, and then abnormal sensory, cognitive, behavioural, and adaptive functioning in children with ASD [8]"
I suggest going through the references carefully to see if they support the statements as a primary source of evidence and fixing other similar issues that you might identify.
Response 3: Thank you for your comment, we checked the literature in the article, the Chinese paper was cited mainly because it fits the point in the article, especially lines 50-52, which focused on the explanation that high BMI can have some negative effects on children with ASD, which may have been inaccurately expressed, and I have removed this sentence based on a review of the literature.
Thank you and all the reviewers for the kind advice. We cherish this hard-won opportunity and make serious changes based on the comments reviewers have raised. Should you have any questions, please contact us without hesitate. I prefer communication by email and I’m looking forward to hearing from you soon.
